# Where Is Ethology Heading? An Invitation for Collective Metadisciplinary Discussion

**DOI:** 10.3390/ani11092520

**Published:** 2021-08-27

**Authors:** Nereida Bueno-Guerra

**Affiliations:** Department of Psychology, Pontifical Comillas University, 28049 Madrid, Spain; nbguerra@comillas.edu

**Keywords:** Ethology, discipline, philosophy of science, Umwelt, networking

## Abstract

**Simple Summary:**

I analyzed the current state of Ethology (i.e., the study of animal behavior and cognition) from the researchers’ perspective through an online questionnaire that was responded to by almost a hundred participants. Despite that the number of the respondents was small, it is interesting to highlight some of the results, since they align with some published analyses. First, not many basic concepts of the discipline, nor its denomination, reached homogeneous consensus. This is alarming because researchers need common vocabulary to communicate effectively between them. Second, despite the enormous biodiversity existent, the researchers seem to be biased towards studying primates, our own family species. Also, the apparatuses employed in the studies are not always adapted to the species’ specific characteristics, so our conclusions about their behavior or cognition might be biased. Fortunately, the technology needed to conduct further studies already exists. However, there is not enough funding or collaboration with engineers to make it real. Establishing big scale networking, adopting some research principles such as transparency, and broadening gender and ethnic diversity in research teams may help in adopting new endeavors.

**Abstract:**

Many factors can impact the advancement of scientific disciplines. In the study of animal behavior and cognition (i.e., Ethology), a lack of consensus about definitions or the emergence of some current events and inventions, among other aspects, may challenge the discipline’s grounds within the next decades. A collective metadisciplinary discussion may help in envisioning the future to come. For that purpose, I elaborated an online questionnaire about the level of consensus and the researchers’ ways of doing in seven areas: Discipline name and concepts, species, Umwelt, technology, data, networking, and the impact of sociocultural and ecological factors. I recruited the opinion of almost a hundred of colleagues worldwide (*N* = 98), both junior and seniors, working both in the wild and in the lab. While the results were pitted against the literature, general conclusions should be taken with caution and considered as a first attempt in exploring the state of the discipline from the researchers’ perspective: There is no unanimity for the discipline’s name; 71.4% of the researchers reported there is limited consensus in the definition of relevant concepts (i.e., culture, cognition); primate species still predominate in publications whereas the species selection criteria is sometimes based on fascination, chance, or funding opportunities rather than on biocentric questions; 56.1% of the apparatuses employed do not resemble species’ ecological problems, and current tech needs would be solved by fostering collaboration with engineers. Finally, embracing the Open Science paradigm, supporting networking efforts, and promoting diversity in research teams may help in gathering further knowledge in the area. Some suggestions are proposed to overcome the aforementioned problems in this contemporary analysis of our discipline.

## 1. Introduction

Any ethologist will be familiar with how the four whys of Niko Tinbergen [1] revolutionized the questions and designs that ethologists could use. Indeed, the implications of his thoughts together with those of Konrad Lorenz and Carl von Frisch positioned Ethology at the highest level of human scientific recognition, which earned the Nobel Prize in 1973. This social event became a turning point for the area. Even the laurates would have never expected such award: “Many of us have been surprised at the unconventional decision of the Nobel Foundation to award this year’s prize ‘for Physiology or Medicine’ to three men who had until recently been regarded as ‘mere animal watchers’”, said Tinbergen in his Nobel Lecture [1]. Lorenz, during the banquette, reflected on the consequences of the award saying that “this trust goes so far that under certain circumstances world opinion about the importance of an entire branch of research can be influenced by this judgment” [2]. They were correct: The Nobel Prize established the discipline; new vocations were attracted; departments flourished across continents, and research grew. However, almost half of a century has gone by since then and new concepts, theories, methodologies, discoveries, inventions, and events have happened. Far from being viewed as threatening, these new knowledge and tools can nurture and empower the potential and the extent of disciplines [3]. How has Ethology embraced them? Has Ethology been consolidated as a discipline since the Nobel Prize? What are the further challenges to come? As one prominent ethologist stated, it is never a bad time to invite one to rethink the scope and the basic principles of animal behavior [4]. Therefore, the purpose of the present manuscript is to provide a collective look into Ethology by ethologists to produce an updated “metadisciplinary analysis” that may help in building a common and solid ground for the discipline’s advancement.

To provide a broad analysis about the discipline but still produce a feasible abbreviated piece of research, the present text has been limited to seven areas: Concepts, species, Umwelt, technology, data, networking, and future. The selection of these areas has been grounded on three considerations: the principles of philosophy of science on how disciplines are consolidated, see [5]; the fact that new discoveries and inventions impact disciplines [3]; and the areas that previous analyses of Ethology have investigated (e.g., [6,7,8]). Next, the reasons that these seven areas were selected are detailed.
Concepts: Each discipline receives a concrete name to be addressed in the United Nations Educational, Scientific and Cultural Organization (UNESCO) Nomenclature coding [9]. This name should be shared by researchers to demarcate the area of knowledge from others. If researchers choose different names for the discipline, eventually new methodologies, new experimental designs, new departments, working positions, and differentiated disciplines arise [10]. In the case of Ethology, Shettleworth [7] (Figure 4, p. 215) composed a picture of how there were different names given to the study of animal behavior and how they intersected each other. It is therefore timely to explore whether researchers nowadays find consensus on the name assigned to the discipline or not. Moreover, any given discipline is grounded on a body of theoretical concepts and assumptions shared by all the researchers, so comparisons and reviews can be done (see the Handbook of Comparative Psychology [11]). However, this has not always been the case of Ethology. To name two prominent examples, Levitis, Lidicker, and Freund [8] showed how there was no unanimous definition of “behavior” between behavioral biologists and a debate on how “cognition” should be framed has largely existed in Ethology (e.g., [7,12]). Moreover, the findings of other disciplines, such as the coined “plant intelligence” [13] or new techniques such as the genome editing CRISPR-Cas9 (Clustered Regularly Interspaced Short Palindromic Repeats and CRISPR-associated protein 9) [14] may challenge core ethologic concepts such as “intelligence” or “species”. Therefore, it is timely to explore which concepts find more or less consensus in the discipline to apply for common agreements, if needed.Species: Any consolidated discipline has some specific object of study, which in the case of Ethology is (the behavior of) species. However, the question of which species should be studied and the reasons to study them is unsolved (e.g., [6,7,15,16]). As certain species selection criteria might bias the knowledge of the discipline (i.e., anthropomorphism can lead researchers to selectively study non-human primates over other species), for a discussion about the future of the discipline, it is interesting to explore the species that researchers are currently studying and their reasons to select them.Umwelt: Disciplines possess specific methodologies adapted to their object of study. In the case of Ethology, this is of special relevance because some animals (i.e., humans), with their species-specific perceptual and motor system (known as “Umwelt”, [17]), are studying other animals (i.e., chosen species), with potentially different species-specific systems. Therefore, caution to employ ecologically valid apparatus and to consider the different factors that affects each species’ Umwelt in the experimental designs is crucial [18,19]. Exploring whether the Umwelt of the species is being consistently considered by researchers in their experiments would reveal the health of the discipline.Technology: Inventions can revolutionize an area of research [3], as it happened with the development of the microscope for different disciplines. In Ethology, the development of new devices, such as drones, OMICS (an abbreviation for all the biological disciplines whose names end in the suffix -omics, such as genomics), or neurobiology measurement apparatus, as well as the release of new machine techniques, such as big data analysis, may challenge how data are collected and which new questions the discipline can ask. Knowing which technology is currently used or missed may help to envisage the future methods of Ethology.Data: All scientific areas eventually provide data that need to be analyzed and published to become part of the discipline knowledge. However, the analysis techniques are not alien to fashions and the publication system is sometimes corrupted by how humans configure scientific paths (i.e., publish or perish) or disregard ethical principles (i.e., see discussions about the replication crisis in [20,21]). Getting to know how data are being analyzed in Ethology and how the different crises are being solved may illuminate the modifications to be done for the advancement of the discipline.Networking: A decade ago, it was stated that “people publishing on comparative cognition are not just talking to each other” ([7], p. 212). Currently, within a progressively interconnected and globalized world, networking is easier than ever and collective collaboration might become the seed of future robust data. Therefore, analyzing whether researchers do engage in collaborative research and maintain sporadic contact with other colleagues is a first step to explore whether the 2009 statement is still accurate.Future: Many environmental, cultural, and societal changes may permeate Ethology and affect the future of the discipline: Universal events such as climate change, the effects of human action over the environment (known as “Anthropocene”), the inclusion of ethnic minorities and queer diversity in research teams, or the reconsideration of ethical questions about animal welfare, to name a few. Starting a discussion about which of these changes are deemed as relevant by current researchers may help to understand how Ethology is preparing for the new years to come.

Finally, to make this analysis “metadisciplinary”, the examinees of the aforementioned areas should be ethologists since their daily practice entails working with them. Therefore, the opinions of almost a hundred colleagues worldwide were collected (*N* = 98, from at least 40 different institutions and 14 nationalities) in which both junior and senior researchers participated. Among the latter, many of the currently well-known and leading researchers responded, so that both novice and experts could contribute their vision to this discussion. It is important to note that even though the total number of researchers devoted to Ethology is difficult to estimate, the recruited sample is unlikely to be representative enough of the whole professional group. Hence, the conclusions of this study must be taken with caution. However, this analysis can serve as the starting signal of a collective contemporary analysis of our discipline that may foster further fine-grained studies. Next, I will provide the characteristics of the sample to then go one by one for each of the seven areas providing a brief introduction, sharing the results of the sample’s opinion, and discussing the main conclusions.

## 2. Materials and Methods

Two semi-structured questionnaires with questions regarding the seven areas mentioned above were distributed online. Both questionnaires were divided into six sections (see the full list of questions in the annex of Appendix A): (1) “Demographic data”: Continent of origin, continent of their academic affiliation, age (optional), years of experience, type of worksite (wild, free-range or lab), e-mail (optional); (2) “Our discipline”: Name given to the discipline, level of consensus about some theoretical concepts; (3) “Species”: The current species the researchers study versus the desired species to study, criteria for the species selection; (4) “Procedure”: Awareness about the meaning of Umwelt; use of ecological apparatuses; the level of relevance attributed to some species-specific characteristics in any experimental design; the current and the potential uses of technology both in the field and in the lab; the statistical analysis techniques employed; (5) “Data Analysis”: The level of impact assigned of some social, environmental, ethical, and academic factors to our area of study; and (6) “Past, present and future of our discipline”: Assessment of some historical moments that contributed to the area or potential future challenges that the discipline will face and how much networking they did.

The first questionnaire was exploratory and mainly contained open questions (e.g., “How would you name our discipline?”). It served to extract the most repeated responses to configure a second questionnaire, which mainly contained multiple-choice questions (e.g., “How would you name our discipline? Please, choose only one even when many are related to each other or even if there is none that entirely conveys your preference”). The second questionnaire can be accessed here (shorturl.at/uKLRT accessed on 8 August 2021) and the full list of questions and responses can be checked in the Annex. Responses from both could be pooled after coding (see data sheet attached in Appendix A, which can also be downloaded from the author’s institutional repository here: http://hdl.handle.net/11531/59820, permanent link accessed on 8 August 2021). Both questionnaires were elaborated in Google Forms and participants provided written consent to participate. Only some questions were mandatory to avoid participants’ frustration or abandonment, therefore, in the Result sections, the total number of respondents will slightly vary between the analyzed variables.

The questionnaire was distributed between February 2021 and March 2021 among the authors’ colleagues both by email and text phone services (mainly primatologists); among the collaborators of a collective book about different species’ Umwelt [19], who were mainly top-referent researchers of a wide array of animal taxa (e.g., ants, birds, and hyenas, to name a few); among different associations for the study of animal behavior (e.g., the Association for the Study of Animal Behaviour (ASAB), the Australasian Society for the Study of Animal Behaviour (ASSAB)); and among colleagues with active profiles in Twitter, asking all of them for dissemination. In an effort to avoid some potential Western bias, some overseas institutions (e.g., Kyoto University; Macquarie University), associations, and non-governmental organizations located in Africa, Asia, and Oceania working in the lab/in the field were also contacted. Therefore, the data collection technique was a non-probabilistic snowball method.

## 3. Sample

Sociodemographic: The sample is composed by 98 participants (round 1: 63, round 2: 35). Gender is not indicated because it was not included as a mandatory question. Despite the effort put in including underrepresented minorities, the sample nationality at birth (first data in the parenthesis) as well as the current institution of origin (second data in the parenthesis) were mainly from WEIRD (Western, educated, industrialized, rich and democratic) societies [22]: European (66.7%, *n* = 58; 69.4%, *n* = 68), and North American (24.1%, *n* = 21; 20.4%, *n* = 20). There was very little representation from South America, Asia, and Oceania (9.1%, *n* = 8; 10.2%, *n* = 10).

Education: Half of the participants have studied some Biology degree (55.1%, *n* = 54), followed by Psychology (24.5%, *n* = 24). Other studies were also reported, such as Philosophy, Anthropology, or Environmental Ecology.

Age and professional experience: Only 79 participants reported their age, which ranged from 21 to 79 years old with a mean of 39.8 years (*SD* = 13.25). With regards to professional experience, the sample is almost equally distributed among all levels. There is almost a quarter each of junior professionals (0 to 5 years of experience, 23.5%), young leaders (5 to 10 years, 31.6%), senior professionals (10 to 20 years, 21.4%), and experts (more than 20 years, 23.5%). For analysis purposes, the variable “professional experience” was dichotomized by pooling those researchers with 10 or less years of experience, called “juniors” and those with more than 10 years of experience, called “seniors”.

Worksite: I considered three different worksites depending on the level of freedom that the studied animal had (lab, free-range, wild). From the total sample, the majority work at labs/zoos (45.9%, *n* = 45) or combine free-range/wild studies with experiments at labs/zoos (32.6%, *n* = 32). Only 21.5% of the sample works exclusively with non-human animals in their natural habitat or at natural reserves. For analysis purposes, the variable “worksite” was dichotomized into “lab” and “field”, and it was coded as “lab” for those participants that reported working in the lab and those working in the lab and in free-range worksites and coded as “field” for those participants that reported working in the wild and those working in the wild and in free-range worksites. Those who marked all options (*n* = 16) were excluded.

## 4. Results

### 4.1. Concepts



*“The scientist does not study nature because it is useful; he studies it because he delights in it, and he delights in it because it is beautiful. If nature were not beautiful, it would not be worth knowing, and if nature were not worth knowing, life would not be worth living”*
(Henri Poincaré)


#### 4.1.1. Discipline Name

All disciplines become renowned as scientific once they have a common name to be referred to and once there is some common pool of concepts that constitute the core of their set of research questions and studies [10]. Indeed, the six-digit UNESCO nomenclature does make an effort to nominate and separate all the different disciplines by a single name and code. This way, not only is the discipline and a particular “Universe corner” [23] recognized and differentiated among others, but also specific departments, positions, and funding options can emerge and consolidate its expansion [10]. Along the history of Science, many disciplines have undergone crucial denomination changes that led to the emergence of different methodologies and new ways of understanding concepts (e.g., Genetics [24]; Physics [25]). Until recently, there have been several definitions for the discipline whose central interest is getting to know more about the behavior and cognition of non-human (and human) animals [26], but the word “Ethology” has served as the wider umbrella term to embrace all. However, after the progressive emergence of different associations, conferences, and handbooks that chose related but different denominations to refer to this area of study, and after the emergence of the cognitive revolution during the 1960s, it may have happened that not all researchers would pick up the same name for the discipline (for the evolution of the discipline denomination, the interested reader may refer to elsewhere, e.g., [27]), with a subsequent risk to its consolidation. Thus, the participants were asked which preferred denomination would better embrace the studies they do. The response options included the terms of the famous Hinde’s title [4] (Animal Behavior, Ethology, Comparative Psychology) plus two other terms (Animal Cognition and Behavioral Ecology), thus covering the names of some of the most reputed scientific journals in the field [6,7], and still limiting the responses to a few options.

The preferred denomination for the discipline among the whole sample (*N* = 98) was Animal Behavior (48%, *n* = 47), followed by Animal Cognition (26.5%, *n* = 26). Other less selected denominations included Ethology (13.3%, *n* = 13), Comparative Psychology (*n* = 8), Primatology (*n* = 1), or Behavioral ecology (*n* = 1), while two instances reported just “other”. If we compare these results to the three UNESCO nomenclatures that refer to studies about non-human behavior (i.e., Ethology, Animal Behavior, and Comparative Psychology), we find that the one with higher level of analysis, namely, Ethology (2408), is not the most chosen one by this sample, but a lower-level area of analysis, namely, Animal Behavior (240102). This choice is relevant for considering the areas of research that grant applicants can be assigned to, for choosing journal denominations, and for organizing academic departments, since it seems that the term Ethology is becoming *démodé*. Potentially, the prevalence of English as the scientific communication language is replacing the Greek etymological origin (*ethos-*, behaviour; *-logos*, study) of the term. Adopting “animal behavior” as the new term for the discipline may be the solution, although it may feel awkward for cognitive scientists (because of the traditional behavioralist vs. cognitivist perspectives) and may not intuitively include humans in the discipline’s scope (because of the traditional differentiation between human vs. non-human animals). Until that new term is widely agreed, Ethology was still used in the title of the present text for historical reasons. In the future, the scientific community should be open to find agreed-upon terms to refer to this area of study (and to potential subareas of study) since many factors crucial for its survival depend on that: Education offer, funding resources, academic positions, and platforms for knowledge dissemination.

The name of any discipline is related to the research questions that researchers pose, therefore the participants were also asked which underlying scientific interest they had when doing their research, providing three response options: Non-human species focus (“To know more about X non-human species”), non-human species comparison (“To know more about the differences/similarities between X and Y non-human species”), and human species focus (“To know more about the differences/similarities between X non-human species and humans”). Most researchers reported to have a non-human species focus (45%). The rest of the sample was similarly divided into a two non-human species comparison (28.6%) and human species focus (26.5%).

At least according to this sample, it seems that Ethology is not biased by some potential anthropocentric vision (namely, a majority of human species focus), but an intrinsic interest in getting to know the nature of non-human species prevails. Interestingly, whereas this vision was shared similarly between lab and field researchers (*N* = 87, *U* = 568.500, *p* = 0.189), the underlying researchers’ motivation varied depending on the denomination they assigned to the discipline (*χ*^2^ = 15.460, df = 6, *p* = 0.017), as it can be seen in Table 1. Concretely, those who chose “Ethology” rarely conducted comparative studies to indagate more about humans and focused instead on non-human species *per se*, both individually or in comparison with another non-human species. Those who preferred the denomination “Animal Behavior” and “Animal Cognition” mostly preferred to focus on one non-human species, although many others were similarly divided between the other motivations. Finally, those who chose “Comparative Psychology” were mostly human-centered.

These results show that, despite that all the researchers felt called to participate in the present study because they did ethological research, they were not unanimous in how to refer to the discipline. Indeed, there are different names for the specialized journals, conferences, or associations (e.g., Comparative Cognition Society, Association for the Study of Animal Behavior, Animal Behavior Society, The International Society for Behavioral Ecology). Each denomination may attract different research motivations and, by extension, different methodologies, topics, funding sources, and researchers, with the potential risk of future disaggregation. The negative consequences of this may be ignoring much of the content associated with each discipline term [28], failing to compose a full picture about the common object of study, which is describing animal behavior and cognition to understand how they evolved. It seems therefore urgent to reach common consensus on how to refer to this field of study.

#### 4.1.2. Theoretical Concepts

The next step for the consolidation of a discipline is harboring a common pool of agreed-upon terms to pull in the same direction and to favor dialogue between colleagues, otherwise the risk of diverging in conclusions as well as precluding comparisons or systematic reviews is very high. The proper use of terms can even radically change the perspective over a discipline, as it happened with the Avian Brain Nomenclature Consortium: “names have a powerful influence on the experiments we do and the way in which we think. For this reason, and in the light of new evidence about the function and evolution of the vertebrate brain, an international consortium of neuroscientists has reconsidered the traditional, 100-year-old terminology that is used to describe the avian cerebrum” ([29] p. 151).

Indeed, the “accuracy” in terminology was deemed by Kuhn [30] as one of the most relevant characteristics for good scientific theories, therefore a recurrent source of scientific conflict is the difference in the interpretation of the words that we use [31]. However, Ethology deals with many concepts in which, despite being covered in a single and accepted word, their interpretation is subject to controversy: Levitis, Lidicker, and Freund [8] investigated whether “behavior” had a common definition between researchers and found that “there was not a single question (…) that produced a unanimous consensus” (p. 107); consciousness (e.g., [32,33]) is said not to have an “universally accepted definition for the term” ([34], p. 210); the same happens with personality (e.g., [35]). It is true that Science advancement does not necessarily require entire homogeneity, but some “degree of consensus is the key” ([36], p. 260). Consensus is defined as “to have a working definition of behaviour that will be as operational and as essential as possible, thereby providing conceptual guidance as to where the boundaries are” ([8], p. 107). To explore the degree of consensus that participants had in core concepts in Ethology, they were firstly asked how much general level of consensus existed in their opinion in the discipline in general, and secondly, what level of consensus they attributed to some concrete concepts.

The majority of the sample reported that there is consensus only in a limited nucleus of theoretical concepts (71.4%, *n* = 70), and this opinion did not seem to be associated with the experience of the researchers (*N* = 98, *U* = 1165.500, *p* = 0.839) nor with the worksite where the participants conducted their studies (*N* = 98, *U* = 636.000, *p* = 0.457). The actual absence of consensus or at least the perception of a lack of consensus that this sample reported should be warning that Ethology might be suffering from some deep conceptual crisis. If these data were representative of the current total amount of ethologists, the conceptual crisis should be directly addressed with collective collaboration before we continue producing much data to avoid establishing little, separate, and independent knowledge.

Indeed, the consensus on core concepts is not homogeneous either, according to this sample. During the first round, there was an open question included about which concepts would find less or more consensus in the discipline. From the responses (*N* = 63), the concepts with higher mode were extracted, yielding eleven items: Cognition, social learning, associative learning, Tinbergen’s questions, Evolution Theory, culture, tool use, emotion, linguistics, cooperation, and prosociality. In the second round, the researchers (*N* = 35) were asked about the degree of consensus on these eleven concepts using a Likert scale (ranging from 1 = little consensus to 3 = lot of consensus), see Table 2 for the results. Importantly, I ran two cross tables to explore the distribution of the responses according to the researchers’ worksite and according to their years of experience, but neither factor yielded any significant differences.

The theoretical concepts that showed more consensus between the researchers were the principles of the Evolution Theory (*M* = 2.80, *SD* = 0.473), the Tinbergen’s questions (*M* = 2.77, *SD* = 0.490), and the definition of associative learning (*M* = 2.63, *SD* = 0.598). It seems that the discipline foundations are clearly established, or at least are deemed as commonly agreed. If this was true, it is good news because it provides solid soil to build new knowledge. Indeed, it is informative about the initial origins and emergence of our discipline (i.e., Darwin, Skinner, Tinbergen). The common ground of Ethology seems to be the fact that there are selective ecological, sexual, and interspecies pressures that affect the development of species; that there are four approaches to study animal behavior: Adaption, phylogeny, mechanism, ontogeny; and that behavior can be promoted or made extinct through reinforcement and punishments. Yet, the potential existence of consensus does not exclude the emergence of some integrative updates proposed in light of new findings and techniques. For example, on the 50th anniversary of Tinbergen’s questions, Bateson and Laland [37] reviewed the relevance of Tinbergen’s seminal legacy for Ethology and proposed to incorporate three more questions, such as “What do we currently understand as causation and function?”, “How is the behaviour inherited?”, and “How can the four questions be integrated?”.

The theoretical concepts that reached only moderate consensus were those related to behaviors that entail more than one individual, such as social learning (*M* = 2.20, *SD* = 0.584), cooperation (*M* = 2.14, *SD* = 0.550), or prosociality (*M* = 2.03, *SD* = 0.664). Perhaps the level of mechanics of the researcher together with the ethogram of the species and how the concepts are operationalized in a given experiment are some factors that start distancing colleagues. The concepts that reached the least consensus were abstract entities such as cognition (*M* = 1.94, *SD* = 0.639), culture (*M* = 1.77, *SD* = 0.547), language (*M* = 1.74, *SD* = 0.657), and emotion (*M* = 1.46, *SD* = 0.611). These results seem representative of the current debates in the area. For example, Bräuer and colleagues [12] recently discussed how cognition should be reframed in Ethology to lose anthropocentrism and gain a more biocentric approach. However, even when it is conceivable that different interpretations of abstract concepts exist, we should consider facing this disagreement because these concepts lie at the frontier with other sister areas of knowledge, such as Psychology, Anthropology, Linguistics, or Artificial Intelligence, to name a few. Whereas this proximate distance between disciplines could be very fruitful in terms of interdisciplinary collaboration, it can also turn to become very dangerous if researchers, not finding a consensus in their own discipline, opted to arbitrarily borrow the formulations of other disciplines, thus adding even more conceptual and theoretical differences to the current disagreement. Tackling these conceptual differences is also very relevant for the discipline growth, because, as already anticipated in the introduction, new findings such as what is coined “plant intelligence” are challenging our traditional understanding of what underlying mechanisms favor the existence of cognition, which eventually has profound consequences in exploring how these capacities evolved.

Probably forecasting these barriers, some researchers already suggested in the final open question of the questionnaire the need for an “international colloquium to agree on definitions” or some “fertile cross-talk”. Similar initiatives have already been done in the shape of special issues, conferences, or forums. Currently, the potential solution for this lack of consensus might be convoking some sort of World Consensus Conference to discuss different concepts, similar to what the Avian Brain Nomenclature Consortium did [29]. The present study could inspire a precise list of the concepts we should start with. One source of inspiration for how this world consensus meeting could be shaped might be the Strüngmann Forum Reports (e.g., [38] especially p. 67-on about social cognition). These are international periodical forums in which some hot topic is chosen, some concrete questions are formulated, and a considerable number of researchers with different perspectives and theoretical models meet for discussion to reach conclusions. Following this map route, but also opening the discussion to non-leading researchers as well as trying to favor the representation of minorities, gender, worksites, and years of experience in the participants, might be very fruitful. Otherwise, the vision of the discipline might be biased again. However, one of the main biases that Ethology is currently facing is found in the species we select to study.

### 4.2. Species



*“There is grandeur in this view of life… and that, whilst this planet has gone cycling on according to the fixed law of gravity, from so simple a beginning endless forms most beautiful and most wonderful have been and are being evolved”*
(Charles Darwin)


If we study non-human animals, which species should we investigate to produce knowledge in the discipline? While the better integrative answer should be all of them, it is true that we lack enough resources (e.g., funding, staff, time, etc.) to do so. Therefore, we need to be selective to produce data and choose only an extremely narrow window of species within the current rich diversity. However, this *a priori* selection is already skewing the real diverse natural world we live in. Some authors have analyzed the species studied through looking at the research published in different relevant journals (see Figures 1 and 2 in [6]; Figures 1–3 in [7]; Figures 1–3 in [39]; Table 1 in [40]). They all conclude that Ethology started focusing on a poor diversity of species (i.e., albino rat, pigeon) to progressively incorporate more different taxa. However, the current predominance of non-human primates in publications is higher than other species: “the species represented shifted toward apes, monkeys, and humans” ([7], p. 212); “the reporting data about primates quadrupl[ed]” ([40], p. 1, see also Table 1 in p. 2). Cronin and colleagues ([41], abstract) also found that, at least in social cognition “while a wide range of species were studied, they were not equally represented, with 19% of the publications reporting data for chimpanzees”. This primate focus to investigate social cognition was also detected by others some years before [42] and the specific predominance of chimpanzees within the studies with non-human primates led to the so-called “chimpocentrism” [15,43]. This might be informative of the anthropocentric bias underlying the species selection criteria of researchers, but other biases can also coexist: “a mixture of taxonomic prejudices, cultural aspects of behavioural ecology as a field, and of academia in general” ([39], abstract).

Therefore, getting to know the species studied and the selection criteria is essential because a potentially narrow-biased vision would be contributing to the foundations of our discipline. Thus, first, participants were asked which group of animals they were currently studying and why. Second, they were asked whether they would be interested in incorporating another group of animals in their research and, being affirmative, which group of animals would be and why. Six potential species selection criteria were provided: Model species for a research question, availability (including low-cost maintenance or easy access to big sample sizes), chance (including job opportunity, funding), fascination about the species, the species being understudied, or the species being “very skilled”. Whereas the first reason would be grounded on answering some specific theoretical question, the existent of model species could also reduce the study of other species. “Availability” refers to the easiness of access and maintenance of the animal, thus skewing the study of other animals such as those in extreme conditions, whose study could illuminate much about how ecological pressures influence behavior or cognition. “Chance” would imply an opportunistic approach to Ethology, sustained by the direction that funding organisms already had chosen for the discipline, but that could also eventually turn into research vocation and generate new research questions. “Fascination” is a subjective reason, grounded in the particular interest of each researcher, which can eventually cause them to look to peculiar species. The “understudied species” reason may reflect an interest to expand our knowledge to other taxa. Finally, the reason framed as “the species being skilled” raises concern about whether “skilled” is a hidden anthropomorphism or, by contrast, whether “skilled” refers to explore non-human capacities that can expand our knowledge on how those capacities evolved.

The participants mostly worked with non-human primates (27.6%), followed by birds (21.4%), invertebrates (11.2%), dogs/wolves (9.2%), and rodents (7.1%) (see Figure 1). There was a minority of researchers that worked with non-avian reptiles, fish or marine animals, wild carnivores, and farm animals (i.e., horses, pigs, chickens). This sample seems to be intuitively representative of the total amount of publications in our area as stated above; however, it could also be providing a biased picture of the full discipline due to the non-probabilistic recruitment followed. The main reason that researchers (*N* = 98) reported some species to be eligible for their studies was being a model species to answer some research question (37.8%, *n* = 37). Thus, it seems that most of the researchers purposefully choose one species for a concrete scientific interest. However, the rest of the reasons reported by the participants distributed very similarly between availability, chance, fascination, unique skills, or being understudied. This variety of reasons should be taken with optimism, since its concurrence leads to including species diversity in further studies.

Interestingly, when the participants were asked whether they would like to incorporate some new species to their studies, an overwhelming majority responded affirmatively (86.7%, *n* = 85). The new species to be incorporated continued being non-human primates, birds, and invertebrates but also singular animals appeared (i.e., deer, beavers, Midshipman fish). Probably because of the singularity of some species, the research reasons to study these animals differed. The first position was tied between fascination about the animal (i.e., “I always dream as a child to study elephants and sometime in the future, I’ll do it”) and model species. This shows that, at least in this sample, researchers seem to combine both their scientific and loving interest at work, which may be a welcoming combination to expand our research to understudied taxa. Yet, the consideration of what a model species is should be revisited, since biases can happen (e.g., considering only male rodents as a model species in neuroethology has excluded reproductive behavior and phenotypic diversity to be analyzed [44]) as well as oversimplicity or funding biases (as a subject pointed out: “question-centered research means that researchers are encouraged to simply pick the easiest model organism for their question which encourages taxonomic bias”).

Some collective initiatives have emerged to fight species reductionism in Ethology. One example is studying the same cognitive process across different taxa with the same experimental protocol. As an example, in 2014, dozens of experimenters joined to study self-control in 36 species through two problem-solving tasks [45]. Thanks to this large-scale collaboration, the researchers could provide phylogenetic conclusions such as that absolute brain volume best predicted performance across species. Other initiatives have tried to explore the same taxa by conducting the same experiment across different species closely related, such as ManyPrimates. This is a project that congregates researchers worldwide aiming to investigate primate cognition and the ecological factors affecting its evolution. They do so through “large and diverse samples from a wide range of species” because “primate cognition research suffers from small sample sizes and is often limited to a handful of species, which constrains the evolutionary inferences we can draw” ([46], abstract). It is true that the enterprise is challenging and not all the factors influencing behavior might be controlled (i.e., upbringing, epigenetics, relation with the animal carers). However, so far, they have established an international network of collaboration both in the lab and in the wild studying very different primate species. Together, they select topics of study and design experimental protocols to be applied in all the working sites. With such a collaborative disposition, the diversity of species is guaranteed, as well as the power of their conclusions. Their success has permeated other animal taxa and other similar initiatives, such as ManyDogs and ManyBirds have emerged. It is true that including new species (whether or not from the same taxa) would not always be a panacea. One researcher in the questionnaire warned that “it may difficult the selection of proper reviewers to properly assess the quality of the research and [the] risk of becoming meaningless” is high. However, there is a solution for that, in specifying the Umwelt of whichever species studied in each manuscript by default.

### 4.3. Umwelt



*“Our appreciation of what is important and what is accessory; what is big and what is small; lies on a false judgment, namely a truly anthropomorphic error.”*
(Santiago Ramón y Cajal)


“Umwelt” is a German term that von Uexküll [17] used in Ethology to refer to the perceptual and motor world of each animal: Not all species’ senses and abilities to move are the same so “there is no real world but as many worlds as species” (pp. 92–93). Even when this is obvious, it is difficult to put oneself into another species’ shoes. This difficulty has given path to a fruitful discussion about critical anthropomorphism and how we humans can easily fall into bias errors when studying animal behavior and cognition (e.g., [47,48]; see [49] vs. [50]).

However, this Umweltian effort should guide our experimental designs. Specifically, following Uexküll’s classification [51], we should ensure in our tests that the specific way in which each species perceives stimuli is taken into account (i.e., *Merkwelt*) and that the actions we expect from the animals fall within their natural repertoire (i.e., *Wirkwelt*). Together with these two factors, it was proposed elsewhere to also consider in our tests how individuals of some species interact with conspecifics and the social factors that may influence its behavior and cognition (i.e., *Sozialwelt*) [18]. For instance, some authors have highlighted that “maintaining the social environment that is characteristic of a species’ natural history as much as possible during cognitive testing improves the socio-ecological validity of the research. The increased validity should be especially pronounced for gregarious species and when social cognition is under investigation” ([41], p. 2). Failure to do so might wrongly lead us to misinterpret the species’ capacity to solve problems and therefore bias any conclusion about their cognitive skills. To these three terms (*Merkwelt*, *Wirkwelt*, *Sozialwelt*) has been recently added the acronym STRANGE. This acronym is a parallel of the WEIRD characteristic of human subjects [22] that were raised to avoid generalizing conclusions with under-representative samples. In Ethology, STRANGE stands for Social background; Trappability and self-selection; Rearing history; Acclimation and habituation; Natural changes in responsiveness; Genetic make-up; and Experience [52]. Taking into account the three Umweltian factors plus the new acronym prior to conducting research could much improve the honesty, ecological validity, and cumulative knowledge in our discipline (indeed, the STRANGE framework has been recently adopted by some journals, such as Ethology or PLOSBiology). However, were participants aware of these terms?

Despite being so relevant for our discipline, the term “Umwelt” was unknown for a third of the sample (37.8%, *n* = 37) and its ignorance was not associated with working predominantly in the field or in the lab (*N* = 98, *U* = 617.000, *p* = 0.410). However, interestingly, not being familiar with the term might not imply that researchers do not regularly take the species’ Umwelt into account in their experimental designs but just a by-product of the potential low frequency of use of the term in Ethology (i.e., “I will be transparent and say I did google Umwelt, because it’s not a word that I usually use… It isn’t widely used. [However,] I try my best to take into account that the way my study subjects perceive the world is shaped by their evolutionary history and is different from mine”).

Maybe it is time to bring the term back into fashion to consider whether we are truly taking all aspects into account. Indeed, considering whether the species’ Umwelt was taken into consideration in the methods of some studies may help peer-review processes and replication. One idea is incorporating an Umwelt checklist into the canonical IMRaD paper structure. The checklist should ask, by default, for information about how the specific species’ senses, motor, social, and STRANGE characteristics were taken into account across the study. Furthermore, the discussion should include a mandatory paragraph about how the object of study and the concrete tasks performed emulated the ecological challenges that the studied species usually face. Another idea would be producing Umweltian species-specific checklists, so that they become mandatorily included in each experimental or observational manuscript about that species, just as the ethical statement is also mandatory. Indeed, including standardized checklists in research reports is becoming popular in other disciplines (e.g., quantitative criminology [53]) and has been recently proposed for reviewing processes in general, with special attention to studies on ecology and evolution [54]. The benefits are evident: With this information, the scientific community could not only be convinced about the validity of the experiment, but they could also replicate the methods better, discuss the interpretation of the results better, and review the whole content better, even when it was about an understudied species or the reviewer was not familiar with the animal.

In an effort to start thinking about that potential checklist, a list of Umweltian factors was included in the questionnaire. Umweltian factors are defined as areas with species-specific differences that should be considered in experimental designs to avoid inappropriateness or low ecological validity (for example, an Umweltian factor is “psychological characteristics”, since some species can be very neophobic whereas others can be very bold (i.e., parrots vs. hyenas), and therefore experimental methods should adapt the conditions and materials accordingly). The selected Umweltian factors were sensory modalities, ethogram, social factors, sex differences, psychological characteristics, and apparatus resembling natural challenges. The participants were asked to score (in a Likert scale from 0 = not at all to 3 = a lot) how much relevance they assigned to each one.

Interestingly, not all the Umweltian factors were equally assessed as relevant (*χ*^2^ = 36.828, df = 5, *p* < 0.001) by the sample (*N* = 35 in all factors except for the question about the ecological apparatus, *N* = 98). The least relevant Umweltian factor for this sample was the species’ psychological characteristics (*M* = 2.37, *SD* = 0.645), although this may be species-dependent because some researchers specifically highlighted the need for considering them, particularly those working with hyenas (e.g.,: “We carefully applied knowledge of hyenas… when developing test protocols to encourage/facilitate participation”) and avian species (e.g., “Devised an aviary test that even a shy, conservative, non-problem solving species can pass”). The second least relevant factor for the sample was the use of ecological apparatuses (*M* = 1.97, *SD* = 0.954), with more than half of the participants (56.1%, *n* = 55) answering that they did not use apparatuses made only of natural (non-synthetic) material that reliably resembled the type of problems the species would find in its natural habitat. The justifications they gave were related to being unable to find a novel ecological method (e.g., “We attempt to get at ecologically valid set-ups, but sometimes, you have to do something very unnatural, like get dogs to look at screens”) or the domestic nature of the species (e.g., “the “Umwelt” of dogs is the human environment. Even in free ranging dogs, they feed from human trash and they are familiar with human apparatus”). However, other authors particularly highlighted the relevance of this factor (e.g., “[We sould] Look at the problems elephants actually face and solve in their natural environments and explore those rather than applying artificial tests of cognition used in other species”). Indeed, using artificial (for the species) materials or situations is worrying in terms of ecological validity because we may arrive at unrealistic information since the animal is not used to that artificial environment.

In general, the species’ perceptual system was considered as the most important factor to be taken into account when designing experiments (*M* = 2.91, *SD* = 0.284). The participants provided many illustrative examples about which sensory modality they considered in their studies (e.g., “Avian vision”, “Dogs sense of their own size”, “Goffins are very ‘haptical’ animals, they learn about the world by touch. We cannot rely on [visual stimuli] … even knowing that their vision is good”) and how they particularly adapted the experimental material to it (e.g., “Olfaction is very prevalent so [I] clean props”) or how they needed to specifically code each trial (i.e., “Recording a song which couldn’t be correctly analyzed with the use of only our ears because of it complexity”), thus acknowledging the human’s sensory limitations compared to that of the studied species. The good news is that the rapid development of technology (sometimes coming from the artistic scene, see the “metaperceptual helmets” by Connolly and McKenzie) is allowing us to gain novel access to those specific-species alien worlds different from ours to improve our Umweltian empathy.

### 4.4. Technology



*“We are on the brink of an age when technology will redefine birth, food, sex and death-the fundamental elements of our existence. (…) How much are we about to hand over technology?”*
(Jenny Kleeman)


The evolution of technology, techniques, and new inventions can certainly revolutionize a discipline by allowing to explore something that was imperceptible to human senses (i.e., microscopes provided Biology, Medicine, or Physics the evidence and arguments for different elements such as virus or subatomic particles, e.g., [55]); by modifying rooted assumptions (i.e., genome sequencing contributed with evidence for the settlement of the evolution theory and forced taxonomy to be adjusted); by compiling multiple data, accelerating data processing, and providing opportunities to forecast and prevent events (i.e., deep learning is used to predict wind power, potential new crimes or diseases, e.g., [56]), or by providing new areas of interest that sometimes can collide with ethical issues (i.e., CRISPR method, artificial intelligence). Hence, no discipline can be alien to new discoveries and inventions and should be open to embrace those whose advantages surpassed the drawbacks. Which current available technologies can have an impact over Ethology? Although this question becomes rapidly obsolete, there are illustrative examples of how technology contributed to expand our knowledge in the area.

For instance, the question about whether non-human animals had theory of mind was controversial and produced numerous opposing correspondence between researchers for more than four decades (e.g., [48,57,58]). However, the arrival of eye tracker techniques, which allowed for following the gaze of subjects with a minimal margin error, allowed to incorporate new ways to explore the existence of theory of mind in apes with a simple experimental design based on the anticipatory look on others’ movement [59,60]. Plus, the method is non-invasive and restraint-free [61]. Another milestone of technology in our discipline has been the use of drones in marine studies, because “behavioural observations are typically limited to records of animal surfacings obtained from a horizontal perspective” but “drones UAS provided three times more observational capacity than boat-based observations alone (300 vs. 103 min); provided more and longer observations plus enable documentation of multiple novel gray whale foraging tactics and social events not identified from boat-based observation” ([62], p. 359). To study songbirds in zebra finches, some authors have employed miniature, animal-wearable wireless microphones and brain recording devices [63], being able to conclude the neuronal activity of the birds’ song system and their stack calls. Even scientific dissemination about animal behavior, such as broadcasted documentaries, have benefited from technology by using realistic “animatronic spy creatures” that infiltrated some species’ environments to explore how they expressed complex emotions, such as grief [64].

Do ethologists regularly use technology in their experiments? In the sample, the majority of the participants reported using sophisticated technology (63.3%, *n* = 62) with no differences between working predominantly in the lab or in the field. Plus, the type of technology used did not vary a lot depending on their worksites either. The tech devices or methodologies included in the response options were extracted from the open responses in the first round (*N* = 63): Touchscreen, automatized feeders, artificial chambers, eye tracking, ECG/fMRI/Xray, sound analysis techniques, GPS/Tracking/Telemetry devices, OMICs techniques, drone, devices to overcome human sensory limitations, and computer simulations. The devices to overcome human sensory limitations and the devices to analyze sound were in the top three most frequently used technology both in the lab and in the field. The remaining position differed for obvious reasons (i.e., lab: Artificial chambers, field: GPS tracking and telemetry devices). Researchers were also provided with some blank space to indicate other technology they used, different from the options provided. The technology reported was very attached to the species’ characteristics or to the type of study conducted (i.e., robotic raptors, RFID—PIT (Radio Frequency Identification through Passive Integrated Transponder) tag technology, Electrophysiology rig).

Researchers were also asked about the “dreaming device they would like to have”. Interestingly, their responses did not seem that unfeasible in terms of the current technology possibilities but rather in terms of other factors such as budget, portability or size, power, accuracy, animal wellbeing, and all-proof material (see Table 3). It seems that Ethology would benefit from joining forces with engineers. To do so, some effort should be made to announce our needs within a public collaborative framework, such as some helping forum or specific website/app, this way promoting a new interdisciplinary network with many labor possibilities. Until then, researchers could find ways of collaboration with other colleagues sharing their tricks and affordable ways of doing comparative research (e.g., [19,65]).

According to our current knowledge and possibilities, the true dream technologies reported were artificial intelligence and automated devices that replace the human supervision or the human–computer interaction. This technology is desired because it may “reduce the current workload” and increase “objectivity” by reducing the “human observer” interpretation (see a revision of automated techniques in comparative psychology and affordable alternatives [65]). Some participants suggested to produce an “experienced ethologist robot”, “programmes that could automatically record and identify different behaviours”, and some “automated processing of data”. It is true that robots are increasingly being used in Ethology [66]. However, one subject raised some concern about this potential technology: “the current trend in some areas to try and automate/quantify all behaviour without understanding either the behaviour or the ecology of the animal” is risky, since researchers’ knowledge, adjustment to the research question, reasoning, ability to find errors, or to identify new, interesting interactions are (still) unreplaceable. It seems that we are facing a moment that is comparable to the 1960s’ computer metaphor. Yet, as Powell and Rosenthal remind us [67], the use of artifices will not solve previous assessment difficulties alone, unless there is a rigorous experimental design.

Until then, researchers need to face much more worldly issues, such as the difficulties to afford sophisticated technology. That is why some free or cheap fee-subscription software initiatives specifically addressing ethological purposes have emerged. These programs allow one to conduct intricate experiments and observations (such as detecting and identifying maned wolves, e.g., [68]), but also, free software has evolved the way that researchers process, analyze, and report data.

### 4.5. Data



*“‘That is so,’ replied Diagoras;*

*‘it is because there are nowhere any pictures of those who have been shipwrecked and drowned at sea’”*
(Cicero)


Data are the main raw material upon which human knowledge is built. That is why it becomes relevant to discuss how that data are obtained and reported in each discipline. Recently, Science in general has been subject of the so-called “replication crisis” [69], an inability to reproduce the same data of previous studies, which challenges whether the original data could be considered real facts. Indeed, this crisis has extended to Psychology [70] and Ethology. The Animal Behavior and Cognition journal published a paper last year [71] to reveal some problems in the discipline with relation to data: Replication crisis, p-hacking, statistical analysis preferences, or small sample sizes, that other authors worldwide had also highlighted (e.g., [21,72,73,74]). Concretely, one pre-print entitled “The illusion of science in comparative cognition” became rapidly widespread [20]. The authors acknowledged some difficulties to achieve replicability: “(1) a lack of access to the species of interest; (2) real differences in animal behaviour across sites; and (3) sample size constraints producing very uncertain statistical estimates”. Replication becoming a new daily practice seems not to be appealing, nor “part of our scientific culture” and frequently finds “general disdain by journal editors”, ([75], abstract). However, the crisis, or at least the awareness of the crisis, has reached unprecedented levels, and the solution that authors more strongly endorsed is embracing Open Science: Data sharing; improving the visibility of negative results (so that Diagoras could be informed about the ships that drowned, see [76]); to improve statistical thinking, and to explore new infrastructures to create and combine the data necessary to understand how cognition evolves [20,71]. To see whether researchers in Ethology are inclined to adopt Open Science, the participants were asked about their sample sizes, the statistical program, and the type of analysis they most frequently used and the impact (in a Likert scale ranging from 0 = not much to 3 = a lot) they assigned to different issues related to reporting data: Small sample sizes, the reproducibility crisis, a publish or perish system, and the Open Science paradigm.

#### 4.5.1. Sample Size

In general, the participants were divided into half who conducted studies with more than 20 subjects (49%, *n* = 48) and half whose studies ranged from 1 to 7 subjects (11.2%, *n* = 11) and from 8 to 20 subjects (39.8%, *n* = 39). Taking into account the worksite of the respondents, the studies done in the field (free range or wild) rarely had small samples while studies in the lab varied enormously ranging from a N = 1 sample to more than 40 subjects. The comparison between the group of animals and the number of subjects in the samples only approached significance (*χ*^2^ = 48.883, df = 36, *p* = 0.074). The studies with more than 20 subjects in their samples were mostly comprised by invertebrates (72.72%), birds (52.38%), and non-human primates (40.7%). It is therefore more likely that factors such as management, availability, and the research question itself might account better for the sample size.

#### 4.5.2. Statistical Program

The most preferred software package was R (60.2%, *n* = 59) followed by SPSS (25.5%). The remaining percentage distributed similarly across other software (i.e., Matlab, Python). Being a junior or a senior researcher yielded significant differences in the software choice (*N* = 98, *χ*^2^ = 15.719, df = 2, *p* < 0.001) with juniors being more inclined to use R (77.7%, *n* = 54) than seniors (38.6%, *n* = 44). The arrival of free software also brought the need to learn programming, and it may have happened that senior researchers had more difficulties in incorporating this new skill.

#### 4.5.3. Statistical Analysis

I provided the participants with different statistical analysis options, which entailed different levels of assumptions or control over the causation and prediction of any studied phenomenon to see which they usually perform. The participants could pick up several options from the list: Descriptive, non-parametric, parametric, correlation, logistic regression, GLMM, and Big Data analysis. The preferred type of analysis was GLMM (77.5%, *n* = 76). This shows that our discipline seems very interested in knowing which factors contribute, how much, and how they interact to explain some behavior. The problem is whether we are measuring and introducing all the potentially influential factors in the model. That is why some authors, rather than focusing on how to measure data and how to report the results, highlight pre-data actions, specifically, providing a good hypothesis [77]: “we don’t just want science to be reproducible. Generating better hypotheses is at least as important (…). Who cares if you can replicate an experiment that found that people think the room is hotter after reading a story about nice people? Will this help us to develop better theories? You can craft a fun story about that result, but can you devise the next great scientific question?”.

According to Table 4, Big Data was the least performed analysis (15.30%, *n* = 15). Due to its low frequency, it was interesting to explore where in the discipline it was being used. Concretely, it was performed similarly regardless of the worksite; it was mainly associated with sample sizes comprised of more than 40 subjects (60%, *n* = 9) of non-human primates, birds, rodents, and insect species and it was preferentially executed to compare two non-human species (66.6%, *n* = 10). Potentially, as fashion and technological possibilities influence which statistical analysis researchers usually perform [78], as it happened with GLMM and Bayesian models [78,79], the future may embrace Big Data and Deep Learning analysis in animal research just as they are already being embraced in many different disciplines [80].

#### 4.5.4. Reporting Data

In this section, the participants were asked about the dissemination of the results, in particular, how much relevance they assigned to small sample sizes, the reproducibility crisis, the publish or perish system, and the Open Science paradigm. The results show that all these topics are deemed to have much or a lot of impact in our discipline by the majority of the sample (*N* = 35): publish or perish system (80%), small sample size (65.7%), reproducibility (63%) and Open Science paradigm (43%).

“Publish or perish” was the title of a letter disseminated by Nature in 1962 [81]. Already during the past century, the leading journal reported to receive six times more the number of studies they could publish. In this editorial, the author overtly blamed the academia about the publication system they had nurtured. In his opinion, the system was based more in quantity than in quality, and it had been established for lecturers and researchers to obtain academic positions: “It is also deplorable that a scientist should depend so much on his published works for his own professional advancement. This all tends towards rushing into print, writing of verbose articles and papers, claims for priority sometimes followed by voluminous and even acrimonious argument, little of which adds to the advancement, much less the dignity, of science itself” ([81], p. 709). Strikingly, this opinion seems to be valid more than fifty years later, since Nature published again an Editorial entitled the same [82], albeit this time to warn about the amount of falsified or fabricated data that researchers were producing (and even admitting to produce, see [83,84]) due to this pushing system. The low quality of Science that this system may produce was also highlighted by one of the participants in the sample: “researchers are pushed towards generating publishable data over scientific quality”. That is why currently there are many movements fostering slow science, one example being the permanent manifesto exposed in the frontpage of the Slow Science Society website [85]. Maybe it is time, not only for Ethology, but for researchers in general, to stand for better academic policies that may reinforce not only publications but also fruitful discussions and thoughtful activities, which have always been the seed of Science.

The replication crisis also reached much concern in the sample whereas the Open Science paradigm did not. One constraint that replication finds is not only the high rate of rejection by editors [75], but also not complying with Open Science and not regularly publishing negative results. Indeed, the bias to positive results has been found in all disciplines in association with the so-called Hierarchy of Science, with social and animal sciences being at the top or intermediate levels of the bias [86]. Perhaps researchers, who despite being pursuing truth through Science have a human nature, feel embarrassed to report that their initial ideas were not achieved as they expected. One subject of the sample endorsed this idea adding that some research groups may be so strictly attached to some idea that this could preclude them to conduct certain experiments or disseminate opposite data: “I would say that people are “in love” with their hypothesis. Different labs defend different ideas and design studies explicitly to confirm them, usually over-selling their findings. I miss more objectivity.” However, being frightened to be wrong, is being frightened of one of the main principles of science, falsifiability [87].

Therefore, embracing the Open Science paradigm may help to improve past barriers and shortcomings in the publication system. The Open Science scientific framework might be summarized as understanding that any piece of knowledge is part of humankind and therefore it should be available to everyone. The UNESCO Recommendation definition reads ([9], p. 4): “an umbrella concept that combines various movements and practices aiming to make scientific knowledge, methods, data and evidence freely available and accessible for everyone, increase scientific collaborations and sharing of information for the benefits of science and society, and open the process of scientific knowledge creation and circulation to societal actors beyond the institutionalized scientific community”. To achieve this, some actions we could already take are being transparent in our methods, reporting both positive and negative results, and sharing datasheets with a common format so that systematic reviews and meta-analysis could be eased. To these practices we could also incorporate preregistering articles by default, since some authors demonstrated recently that standard articles reported more positive results than preregistered articles (96% vs. 44%) [88]. A new-born society, SORTEE (Society for Open, Reliable, and Transparent Ecology and Evolutionary biology) is trying to endorse these practices in our field. Perhaps the first step to face the changes to come for Ethology is establishing large-scale collaboration and networking.

### 4.6. Networking



*“Are you nobody, too?/Then there’s a pair of us…”*
(Emily Dickinson)


Discussing how social cognition could be better studied, at a Strüngmann Forum, the participants concluded: “In an ideal world, one useful approach would be to compare multiple species using the same or similar methods used by teams of researchers” ([42], p. 289). However, that is not an easy undertaking for one single person, nor even for one single research team. To conduct good science, we need many people, an authentic “team effort” ([40], p. 2). Indeed, von Frisch already acknowledge the existence of a team behind his Nobel nomination: “The effort of one individual is not sufficient for this. Helpers presented themselves, and I must express my appreciation to them at this time. If one is fortunate in finding capable students of whom many become permanent co-workers and friends, this is one of the most beautiful fruits of scientific work” ([89], p. 86).

This simple idea of connecting many people under the same task was the underlying motto for some networking projects, such as ManyPrimates: To start coordinated large-scale international collaborations. In their first publication together, the authors stated: “Inferring the evolutionary history of cognitive abilities requires large and diverse samples. However, such samples are often beyond the reach of individual researchers or institutions, and studies are often limited to small numbers of species. Consequently, methodological and site-specific-differences across studies can limit comparisons between species. Here we introduce the ManyPrimates project, which addresses these challenges by providing a large-scale collaborative framework for comparative studies in primate cognition” [46]. For similar projects to exist, and to achieve such level of coordination, researchers must be willing to network with peers. However, do researchers regularly interact between themselves?

Strikingly, a quarter of the participants (*n* = 26, 26.5%) reported not interacting at all during the year with other peers and, interestingly, being junior or senior did make a difference in this regard (*N* = 98, *U* = 918.000, *p* < 0.05), with juniors reporting an absence of interaction (33.33%, *n* = 54) more than seniors (18.18%, *n* = 44). Juniors are increasingly becoming digital natives and therefore familiarized with social networks; however, it seems that having access to the media does not imply using the media for academic or scientific purposes. Note that juniors in this sample were researchers with 10 or less years of experience, so it cannot be stated that they were at a very early stage of their career nor that they may be alien to most of the conference or discipline discussions. However, we could take this result as a message to improve collaboration between researchers when they initiate their careers.

Some examples of how to foster networking can be borrowed from other disciplines, such as the PsyResearchList for the study of moral psychology. Dr. Meltem Yucel led the dissemination of a public list of researchers interested in the topic who inscribed themselves on a voluntary basis (see https://www.psychresearchlist.com/moral-psychologists.html, accessed on 8 August 2021). This type of actions eases the act of locating peers with similar interests so further collaboration can arise. Indeed, it is demonstrated that the fresher a team is, the more original and multidisciplinary research they conduct [90] and the better they can face the future challenges to come.

### 4.7. Future



*“I would like to be of help during this decisive but not easy period of your journey,*

*in which you prepare for a direct confrontation with life.”*
(Rita Levi-Montalcini)


The first questionnaire included an open question about what future events may shape the future of Ethology according to the researchers’ opinion. From 53 valid and detailed responses, the issues that had a higher mode were extracted, yielding twelve: COVID-19 pandemic, funding, under-represented minorities, lack of dialogue, lack of interdisciplinarity, lack of collaboration, anthropocentrism, climate change, extinction, welfare ethics, absence of longitudinal data, robots and tech innovations. In the second round (*N* = 35), I asked the participants how much impact they attributed to these twelve issues in Ethology by using a Likert scale (ranging from 1 = low impact to 3 = lot of impact); see Table 5 for the results.

The main current challenges qualified as affecting Ethology “much” and “a lot” according to the participants (*N* = 35) were a lack of funding (85.7%), welfare ethics (77.1%), the COVID-19 pandemic (65.7%), and the absence of longitudinal data (63%). It is especially alarming that Ethology suffers from the exclusion of competitive funding sources, which is worsened by the abundant offer of precarious and seasonal positions. One subject in the sample stated: “It is extremely hard to continue after a master’s or PhD, there are little to no position for early career researchers”. Indeed, the discipline job websites are full of volunteer positions, in which the candidates are expected to pay for their travel costs and are offered scarce maintenance in compensation for their sometimes hard, long, and tiring observational field work. However, field work experience is much appreciated in ulterior job interviews as it is associated with successful training in research techniques. Therefore, if we do not stand against this situation, the precarious volunteer offers will be perpetuated.

Another concern shared by the sample is welfare ethics. Indeed, new knowledge can directly challenge our bioethics protocols, such as finding animal sentience in invertebrates [91]. Moreover, if we acknowledge the Anthropocene term for the geophysical epoch we live in, we may need to identify which human activities are affecting the welfare and the behavior of species (e.g., human noise affecting fish, [92]) to be ethically committed to modify certain habits and also to interpret observations accordingly. An evident consequence of human activities is climate change, which is threatening the survival of many species’ groups to the extent of some alarming situations such as the so-called “insect apocalypse” [93]. Being aware of the potential disappearance of some taxa while being the responsible agents is unacceptable. Any living form deserves to be protected and its loss directly challenges the nature equilibrium plus the object of study of Ethology, which is species. Importantly, welfare ethics should not only be defended by researchers working in the field, but awareness should also be extended to those working in labs.

The factors that the participants considered to impact the discipline less were the use of robots or artificial intelligence (31.4%), low peer diversity of gender or ethnicity (35.1%), and lack of interdisciplinarity (42.9%). However, the absence of under-represented minorities in research teams is currently biasing and impoverishing Science. More diversity of gender and ethnicity in scientific teams has proven to yield more eclectic and integrative studies and even attracting more citations [94,95]. However, the access to the scientific pathway is not always easy for female foreign applicants, at least in higher-paying disciplines and private institutions, where departments usually endorse more Caucasian males in mentoring applications [96]. Indeed, even when the access barrier is overcome, a non-WEIRD ethnicity alone could be one key factor not to be eligible for an NIH (National Institutes of Health) award after controlling for factors such as educational background, country of origin, or previous awards [97]. A similar trend occurs with women’s underrepresentation in Ethology, thus biasing the European narrative (see the Special Issue: A Historical Approach in Animal Behavior, 2020, edited by Zuleyma Tang-Martínez [98]). The underrepresentation of sexual diversity within researchers is also considered a current disadvantage. Their experience of negative emotions at science workplaces in general is high [99], but in Ethology in particular, their exclusion [100] can also imply disregarding potentially interesting perspectives about specific topics (i.e., sexual selection). Under this thought, the ABS (Animal Behavior Society) meeting held in August 2021 included a plenary talk entitled “Different People Ask Different Questions: A Queer Perspective on Studying Behavioural Diversity” given by Prof. Dr. Karen Warkentin. Therefore, Ethology should not only take into account all the scientific specific concerns commented on so far, but also the social, cultural, environmental, and political factors that also have an impact on the future of the discipline.

## 5. General Discussion

Ethology has had great historical momentum. However, “the early Giants are going or gone [and] new Giants are now being established and will lead the field” [101]. Now, it is time to think about the future of the field, and the aim of this paper was raising concern and discussing the most relevant challenges to come in seven areas (concepts, species, Umwelt, technology, data, networking, and future) through the responses of almost a hundred colleagues. First, it seems that there is a lack of consensus with regards to some core concepts in the discipline (and even with the name of the discipline itself), so World Consensus Conferences have been proposed to define a common scientific path and overcome misinterpretations and patched knowledge. Second, the underlying research motivations plus the species selection criteria we follow in our studies should be reconsidered to include less anthropocentric and more biocentric questions [12]. This has led to the third idea, the need to include mandatory checklists of Umweltian factors in published studies, as ethical statements are also mandatory, to ease peer-review processes when determining the ecological validity of the conclusions and to ease further replications. Fourth, the raising of new technology inventions will surely facilitate currently tiring tasks, but it should be accompanied by active human reasoning to provide full-sense knowledge. Plus, current and future tech needs could be easily addressed if collaboration with engineers was fostered through public and private initiatives. Fifth, the current untransparent, biased towards positivity and inaccessible publication system should be progressively replaced with an Open Science paradigm. To do so, preregistering studies, establishing standardized formats for sharing data sheets, losing the fear of contradiction and falsifiability, and substituting the publish or perish system for a more collaborative, fruitful, and slow science process would help. Sixth, networking should be integrated as another scientific principle from the very early stages of the researchers’ careers, not only for the advancement of our discipline, since bigger samples and robust data can be better achieved through large-scale cooperation, but also for achieving a less competitive and more enjoyable scientific environment. Finally, the way climate change is affecting the diversity we study and the fact that some minorities are underrepresented in research teams could directly attack the survival of the subjects we study, the integration of the individuals that study them, and the new perspectives that our discipline needs.

## 6. Conclusions

To all these conclusions, one extra aspect should be included: The need to acknowledge and get knowledge from our predecessors prior to embarking on the future. This will help to ensure we do not take the findings and thoughts that have been already posed as novel ideas [28]. This does not mean to be stuck in the past: We also need to be more open-minded than our predecessors were and try to collaborate more than they could for the advancement of Ethology. Indeed, to make the insights of this essay useful, we should figure out how to educate the undergraduate students that will approach Ethology, so that they could already be prepared for the problems we are envisioning now. Thus, the postgraduate syllabus in Ethology should benefit from including modules about theoretical background on seminal contributions in Ethology, the principles of philosophy of science and Open Science paradigm, discussions on the definition of core theoretical concepts, the concept of Umwelt and its application to experimental designs, statistics programming, an interdisciplinary theoretical background (e.g., [102], p. 17; [103]), and welfare ethics.

Despite the limitations of this manuscript (i.e., sample size, potential underrepresentation of researchers in some animal taxa), it hopefully faithfully represented all the participants’ opinion and may serve as a first boost to indagate more in the current status and future directions of Ethology. Moreover, despite all the shortcomings raised, no pessimistic approach should be established, because this discipline congregates three optimistic adjectives: It is varied, since we study biodiversity each day; it is enduring, since evolution never stops its action; and overall, it is eternal, because “a question answered usually raises new problems, and it would be presumptuous to assume that an end is ever achieved” ([89] p. 86).

## Figures and Tables

**Figure 1 animals-11-02520-f001:**
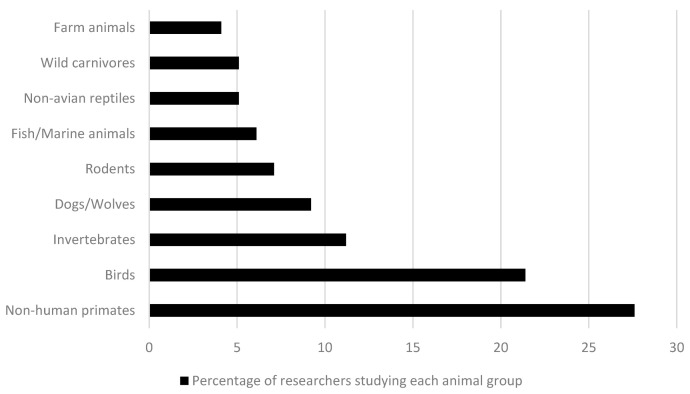
Distribution of the groups of animals studied in a sample of *N* = 98 Ethology researchers.

**Table 1 animals-11-02520-t001:** Matrix with raw data relating the name of preference for the discipline and the underlying research motivation (*N* = 94).

Research Motivation	Ethology	Animal Behaviour	Animal Cognition	Comparative Psychology
Non-human species focus	6	25	12	1
Non-human species comparison	6	12	7	1
Human species focus	1	10	7	6

**Table 2 animals-11-02520-t002:** Level of perceived consensus in eleven relevant concepts in Ethology (obtained from qualitative responses in a previous survey, *N* = 63).

Concept	Min.	Max.	Mean	Standard Deviation
Evolution Theory	1	3	2.80	0.473
Tinbergen’s questions	1	3	2.77	0.490
Associative Learning	1	3	2.63	0.598
Social Learning	1	3	2.20	0.584
Cooperation	1	3	2.14	0.550
Tool use	1	3	2.09	0.658
Prosociality	1	3	2.03	0.664
Cognition	1	3	1.94	0.639
Culture	1	3	1.77	0.547
Linguistics	1	3	1.74	0.657
Emotion	1	3	1.46	0.611

**Table 3 animals-11-02520-t003:** Main barriers to currently dispose more adequate tech devices in Ethology.

Barrier	Device Examples
Budget	Thermal imaging, cheaper micro imaging technology, top notch hydrophones
Portability/Size	Portable DNA sequencer, portable fMRI, GPS tag that could be deployed on <100 g birds, gaze tracking for fishes, wristband date loggers, walkie-talkies in the shape of smartwatches
Power	Higher-speed cameras, tracking multiple animals, tracking small organisms, micro imaging processing power, 3D simultaneously coding animal track and sounds recording, long battery life for microphones and for tags/telemetry—specially to study migrations
Accuracy	Better RFID, GPS satellite tags with high resolution (≤1 m accuracy), better nocturnal cameras
Animal wellbeing	Silent drones
All-proof material	Weather-proof cameras and microphones, GPS telemetry capable of being carried by a wild parrot without being chewed up within 10 min

**Table 4 animals-11-02520-t004:** Most frequent type of statistical analysis conducted by Ethology researchers (*N* = 98).

Analysis	Percentage of Use (*N* = 98)
GLMM	77.55%
Descriptive	67.34%
Non-Parametric	61.2%
Parametric	57.14%
Correlations	55.10%
Logistic Regression	42.85%
Big Data	15.30%

**Table 5 animals-11-02520-t005:** Level of impact that twelve events (obtained from qualitative responses in a previous survey, *N* = 53) may have in our discipline, according to researchers’ opinion (*N* = 35).

Concept	Min.	Max.	Mean	Standard Deviation
Lack of Funding	1	3	2.46	0.741
Welfare Ethics	0	3	1.97	0.954
No Longitudinal Data	0	3	1.83	0.891
COVID-19	0	3	1.77	1.003
Climate Change	0	3	1.63	1.031
Extinction	0	3	1.63	1.114
Lack of Collaboration	0	3	1.60	0.914
Lack of Dialogue	0	3	1.57	0.948
Lack of Interdisciplinarity	0	3	1.54	0.980
Anthropocentrism	0	3	1.46	1.039
Underrepresented Minorities	0	3	1.29	0.926
Robots/Tech Innovations	0	3	1.23	0.910

## Data Availability

The data sheet is included as a Appendix A.

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
