# Peer review of "Where Is Ethology Heading? An Invitation for Collective Metadisciplinary Discussion"

_animals, 2021, doi:10.3390/ani11092520_

Round 1

Reviewer 1 Report

Where is animal research heading? An invitation for collective metadisciplinary discussion

Nereida Bueno-Guerra

The author aims to provide some kind of a review on “animal research” to reveal current “practices” using questionnaire data from N=98 scientists.

I must say it was a very difficult read for me, while I tried to find out: What was the real aim of the author? What was the goal? And also, what kind of a paper this is? Is it a true experimental research paper (with data and statistics), perhaps it is rather an opinion piece or an essay on animal research?

I have two major concerns, but actually three:

  • What is “animal research”? I think the author meant, perhaps, ethology or animal behaviour or comparative psychology or animal cognition, or evolutionary psychology or …. “Animal research, as such, would include research on any aspects of animals, including husbandry, physiology, genetics etc. and not just “behaviour”.
  • N=98 especially if one takes quantitative analysis seriously this N is very low, and very likely non-representative, so for me any outcome for those questions are quite biased, especially because the over-representation of primatologists. Typically, such explorative data collections make sense after N=1000, provided that the sample is representative.
  • Of course researchers may “respond” to such invitations, I do not think that anything like a “world consensus conference” will emerge in the future (L 904). The field is too big, the most questions are to general, the same issues come up in sociology or organic chemistry, and “we” do not agree in many more specific terms like “aggression” or “emotion”. And, there are many other reasons that would be out of scope of this review.

So I am also not sure what I should suggest as a next step. Knowing that the sample is biased and small, most of the “insights” are either trivial or problematic. I think this essay is also too long. Sections can be shortened by taking out individual comments, cutting the Intro and the discussion. I am actually much less optimistic than the author because “behavioural sciences” developed at much slower pace as e.g. genetic. So I suggest a critical re-reading, and perhaps asking older scientists for further suggests how to make this text more interesting.

More specific comments

The question of the discipline name is a nice example for the manuscript as a whole. The whole essay could have been devoted to this issue, if the questionnaire had been designed this way. The approach taken is a naïve because the “name” is just a name and has among others many historical (e.g. the traditional name of the department at the university (e.g. Comparative Psychology), Lorenz’s association with the Nazis), cultural (researcher in the US have not heard the term ethology…) and situational (species/function studied) and linguistic (animal behaviour – is a very descriptive term, like car driver, if anybody wanted a perfect name/label that refers to a serious field of SCIENCE, and they would ask a marketing person then he (would) go for Ethology – ethos+logos) aspect. But most people do not know Greek.

I have also problems with the concepts. What is meant by “theoretic” and “practical” (?) concepts? Is “having an emotion” a theory or a practice? And what it means “having a consensus”? For example, emotion can be defined for very different aspects, depending on the level of investigations, and these definitions would be non-exclusive. Social learning means (for me) learning from the other… what else can it mean? For me “linguistics” is a field of science (see Table), while “tool use” is a very specific issue within “cognition”. So it is very unlikely that scientists agree more on “tool use” than on “cognition”. This outcome is probably also the result of the over-representation of primatologists. For example, see this rather strange idea published recently: https://www.ncbi.nlm.nih.gov/pmc/articles/PMC7838539/

L 311 “If we study non-human animals, which species should we investigate to produce knowledge in the discipline?” I teach my students to focus on phenomena and then chose the species. I my view this is a better strategy. Of course, many researchers get fixed/used to one species, and then use them to answer many questions. This can see in many papers that „inappropriate” species are used for animal studies, e.g. mice in water maze etc.

The concept of “model species” would be also important to mention. There is no chance (for financial and practical and moral reasons) to study “all species”. Just think about the issues of studying all ape species… The latest large effort in Leipzig (by Call and Tomasello and their team) could also achieve only parts of the objectives. Again, this question on the “species” would require a more delicate approach taking into account practical issues of animal welfare, and many moral questions. I am also pessimistic about projects like the MANYPrimates because not only the “Umwelt” matters but also the previous experience of the subjects, including their grantparents’ social upbringing (e.g. cross-generational epigenetic effects) etc. and the caretakers’ behaviour. Standardisation of the variables is not an easy task across research facilities, and who can control how well human assistance actually adhere to the common protocols of testing etc.

Should one study typical pigeons in the lab or go into the “wild” and look for one, two (how many) pigeon species, take them to the lab or set up a field stations… etc. This issues cannot be seriously discussed by answering 2-3 questions, and are very specific to the actual research agenda.

Why did the author ask researchers about “Umwelt” at all? This expression has been avoided (suppressed) in the same way as “ethology”. The use originates from times when ethological conferences (see also International Ethological Conference) were run not only in English but also in German and Dutch or French. One of my former professors in England told me many years ago that he learnt German, especially because he thought German will be the common language for behaviour sciences. But this was 70 years ago, and it did not happen. So we have “environment” instead of “Umwelt”, and can use (or coin) terms such perceptual environment or action environment if we want.

In my opinion it is also irrelevant to ask “in general” what researchers think about the effect of environmental factors. Partly, because, as we know from much experience, the role/effect/influence of these factors is very variable, and there is place for lot of interaction. This is why we normally do statistical analysis. Sex may be important in one situation, and totally irrelevant in another, and there could be an interaction with the species factor etc.

I found the “Data” section quite irrelevant. Findings are trivial, or probably biased because of the small a biased sample used.

Connecting people is the way how science works, but this contacts lead to powerful scientific collaboration when they can be supported by grants etc. Behavioural sciences are typically ignored by many grant agencies, even at the ERC it has been very difficult to find a panel (with experts) for such topics. This also supported by the results in Table 5.

Minor remark:

For me it was very confusing having “N=98” many times, followed by a different %... I would suggest the following reference: (49/98; 50%)

Reviewer 2 Report

I like the idea for the study and I am particularly interested in the extent to which researchers are anthropocentric in their stated aims. I think there is much of value in the data but the paper is lacking in focus. It could be a much shorter, more tightly written paper and I think the important ideas – such as the need to consider an animal’s umwelt – could be emphasized more. There needs to be a stronger opening statement of the goals of the paper and a better integration of the somewhat disparate ideas.

It should be clearer in the introduction why the survey is based around the particular terms chosen by the author and how the survey aligns with the goals of rethinking the discipline. The aims seem a bit loftier than the methods warrant.

What was the reason to ask about the preferred name for the discipline? How was the discipline itself defined? Surely any definition provided would bias the responses.

Although it is impressive to obtain data from almost 100 scientists, this is a relatively small sample for a statistical analysis. If the method is merely descriptive, the author should be more transparent in that goal. However, the author does appear to have used appropriate methods for obtaining a diverse sample.

Years ago, researchers published a paper (in Animal Behavior I believe) that showed little consensus in biologists’ use of the term “behavior.” If the author can find this paper, it would be relevant.

The author should be careful to write not “reach moderate consensus” (e.g. p. 6) but “perceived to have moderate consensus” because she has not determined the actual consensus – just whether the participants perceive there to be consensus. They may wrongfully think that agreement is greater or less than it really is. For example, I don’t think there is great consensus of use of things like evolution or associative learning, although I am not surprised that people might think there is.

 There is a rather abrupt transition between sections, as between sections 1 and 2 (Species).

There is a lot of speculation about why researchers study particular species. The author should make it clear sooner that she also asked the participants for the reasons and tried to determine how much species selection is driven by the theoretical questions of interest. This entire section (2-Species) should be rewritten with a clearer focus on why some biases might exist and whether they might be justified given the questions of interest.

The author might be interested in some articles in the recent commemorative issues of the Journal of Comparative Psychology that document the changes in study species in this journal over different time periods (see papers by Snowdon and Vonk, for example and Fragaszy’s editorial).

Figure 1 is interesting and useful and it depicts an admirable diversity of study subjects among the respondents. However, it is a bit of an odd way to create the categories. What about also presenting species in terms of environment (farm, lab, zoo, pet, wild)? Does Farm animals currently include birds (e.g., ducks, chickens or mostly hoofstock)? What about pet cats or captive carnivores?

The author should also mention the new initiatives ManyDogs and ManyBirds around p. 9.

Burghardt is misspelled on line 434. There are lots of other typos that need correction but the author should at least very carefully check author names.

I think the ideas in the section on Umwelt are very useful and should be highlighted more as an important cautionary lesson for investigators.

Eye-trackers cannot reveal “theory of mind” in nonhumans. These findings are still controversial. The author should be more cautious in her claims between lines 563-568.

What is meant by “no answer is also an answer” is unclear on line 652? It might be better to write “The lack of a clear answer” or “The absence of an answer…”

I find the use of first person in this article distracting. I would not write about “our discipline.” State a proper name for the discipline you refer to. Indeed, it will be important to identify the target discipline clearly from the start – is it ethology, animal behavior, biology???

I think the author should also focus on summarizing the findings, integrating some of her own ideas but not stating that her own opinions should trump those of the majority of her respondents (e.g., lines 877-879).

I also am confused by the number of times the author describes a result and then states “This is why I asked..” or “Therefore, I asked..” It makes it sound like she conducted a follow-up study based on the results of the first study, but there appears to be only a single survey analyzed so I think she should take care to be clear that all questions were already included by a pattern of responses may have led to further probing of other questions.

Line 756, “past vicious”??

The URL on line 823 doesn’t appear to work.

The reference to the “next ABS” meeting is too time-specific. If someone is reading this article ten years from now, this would not read well. State the date of the meeting. Also, is this a “pre-conference” or a “symposium”? It cannot be a conference in a conference as described.

Round 2

Reviewer 1 Report

Comments on the author’s reply („ Where is Ethology heading? An invitation for collective meta-disciplinary discussion”)

First of all, I was very interested to read the long and detailed replies to my comments. I must admit, I wanted to be provocative because otherwise these discussions make little progress. So thanks for being so patient and also for thinking over some of these issues again. I will not answer to each point in detail because I regard the author’s work as a subjective view on this field of biology (I was also a bit surprized to see that the returned ms was already in a proof format and prepared for publication) with a lot of important notions and an optimistic stance that things may and should change, and this is how science should be conducted. These problems are not only restricted to our field but to science in general.

The basic problem remains however, especially because the author puts a lot of emphasis on representative data, data sharing, the crises of repeatability etc., I still wonder how the quantitative aspects of this study can be replicated when the sample is quite biased. However, this may be only my concern.

There are obviously two different types of questions here. Some questions target the scientists’ opinion. Obviously, in this case the use of questionnaires is a good option. Although, even in this case a more balanced sample would have been better. However, if one wants to find out “what scientists actually do”, then there are much better ways for an objective analysis, e.g. by looking at the published papers as a source of such data, e.g. what type of statistics is used, or what species is actually studied. In addition, species bias is more pronounced when one studies ‘cognition’ and much less significant in the case of behavioural ecology.

Actually, I am happy to see that the author writes about the “urge” for consensus. It is an important activity in science, but I think ethology is much further from this possibility than genetics or biochemistry. I am similarly disappointed when I see that the term “ethology” or “animal behaviour” is missing when I have to assign my work into a list of categories within biology because many organisations do not adopt categories put forward by UNESCO, for example this is the case with the ERC.

I am happy to see the change in the title. I totally agree that this branch of science should be called “ethology” but this will be a ‘surprize’ to the majority of the respondents of the questionnaire because half of them suggested ‘animal behaviour’ as a descriptive term. I also doubt that English speaking countries will accept this label instead of “animal behaviour”. Actually, one further support for ethology is that humans are typically excluded from ‘animal behaviour’ which is quite unfortunate.

With regard to the issues of “Umwelt”, I agree that this is an important concept that would deserve a specific treatment, especially because many scientists who study their “favourite” species in the laboratory are not aware of the species’ ‘Umwelt’ in nature. In addition, we know very little about the sensory skills (and perceptual abilities) of most species under study, so much research would be needed that could inform experimental design.  

Small minor issues:

L 36 “the Nobel Prize stablished” SHOULD BE the Nobel Prize established

L 245 N=87, U=.568.500, p=.189 - There seems to be more ’decimal points’ as necessary in the U-value or the value itself is wrong

L 392 I think the text and subsequent citation is a bit contradictory (non-human primates vs …and humans)

non-human primates in publications is higher than other species: “the species represented shifted toward apes, monkeys, and humans”

Author Response

Please, see attachment.

Reviewer 2 Report

I appreciate that the author was receptive to the previous reviews. I think the paper is very interesting and insightful and deserves to be published with some very minor corrections that can occur during proofing.

I appreciate and admire the challenge of writing a scientific paper in a foreign language. There are still some minor errors needing correction. The paper still needs editing by a native English writer. I will not report all of the grammatical issues here.

In the abstract, line 10, the phrase “Far from seen as threatens” does not makes sense. Does the author mean “Far from being viewed as threatening?” Check also line 40.

Line 36 should be “established”  and “They were correct” (rather than true)

Line 147 should be “networking”

The new reference for Tecwyn is missing the first initial of the author – E. I believe.

Lines 177-179, replace the N=98 with the appropriate number for Biology and Psychology or else it is confusing. Similarly, on lines 712-713.

I would avoid the phrase “proper researchers.”

Insert punctuation within quotations.

Data are plural (line 682)

Line 688, replace “society” with “journal” Animal Behavior and Cognition is also misspelled.

Replace “subjects” with “participants” when talking about the human respondents (e.g., line 733).
